

# Lipid accumulation of *Chlorella* sp. TLD6B from the Taklimakan Desert under salt stress

Hong Li[1], Jun Tan[2], Yun Mu[2] and Jianfeng Gao[2]

[1] College of Life Sciences; Key Laboratory of Xinjiang Phytomedicine Resource and Utilization, Ministry of Education, School of Pharmacy, Shihezi University, Shihezi, Xinjiang, China
[2] College of Life Sciences, Shihezi University, Shihezi, Xinjiang, China

## ABSTRACT

*Chlorella* has become an important raw material for biodiesel production in recent years, and *Chlorella* sp. TLD6B, a species with high lipid concentrations and high salt and drought tolerance, has been cultivated on a large scale. To explore the lipid accumulation of *Chlorella* sp. TLD6B and its relationship to external NaCl concentrations, we performed physiological measurements and genome-wide gene expression profiling under different levels of salt stress. *Chlorella* sp. TLD6B was able to tolerate high levels of salt stress (0.8 M NaCl addition). Lipid concentrations initially increased and then decreased as salt stress increased and were highest under the addition of 0.2 M NaCl. Comparative transcriptomic analysis revealed that salt stress enhanced the expression of genes related to sugar metabolism and fatty acid biosynthesis (the *ACCases BC* and *BCCP*, *KAS II*, and *GPDHs* involved in TAG synthesis), thereby promoting lipid accumulation under the addition of 0.2 M NaCl. However, high salinity inhibited cell growth. Expression of three *SADs*, whose encoded products function in unsaturated fatty acid biosynthesis, was up-regulated under high salinity (0.8 M NaCl addition). This research clarifies the relationship between salt tolerance and lipid accumulation and promotes the utilization of *Chlorella* sp. TLD6B.

# INTRODUCTION

Increasing global energy demands cause over-exploitation and shortages of nonrenewable resources such as oil, natural gas, and coal and increase the urgency associated with research on bioenergy. As photoautotrophic organisms (*Schenk et al., 2008*), microalgae are characterized by high lipid concentrations and rapid growth, which makes them the most promising organisms for the development of biodiesel (*Halim, Danquah & Webley, 2012*). The lipid metabolism of microalgae is a complex process that involves the synthesis and metabolism of fatty acids, triglycerides (TAGs), acetyl-CoA, and other compounds, and it is closely related to algal growth and stress tolerance (*Gao et al., 2014*). Environmental changes alter the expression of many genes encoding enzymes in these metabolic pathways and thereby influence microalgal lipid metabolism. Previous studies have found that lipid accumulation in microalgae is affected by changes in the expression of the ACCase subunit

Corresponding author
Jianfeng Gao, jianfengg@shzu.edu.cn

genes *BC*, *BCCP*, and *α-CT*, the fatty acid synthesis genes *SAD*, *KAS II*, *MCAT*, as well as the TAG synthesis genes *GPAT*, *PAP*, *LPATT*, and *DGAT* (*Zalutskaya et al., 2015*; *Huang et al., 2016*). At present, the detailed signaling mechanisms that regulate lipid metabolism in microalgae remain unclear. However, lipid accumulation is known to be regulated by abscisic acid (ABA) and reactive oxygen species (ROS) signals (*Gao et al., 2014*; *Wang et al., 2016*), indicating that microalgal lipid metabolism is closely related to adaptation to unfavorable environments.

*Chlorella* is a eukaryotic, single-celled green microalga that is widely distributed in fresh water, soils, and deserts (*Li, Zhang & Wang, 2016*). It has been widely used in research related to microalgae energy production (*Liang, Sarkany & Cui, 2009*; *Li et al., 2011*; *Liu et al., 2011*). Researchers currently focus primarily on freshwater *Chlorella*, and there have been few studies of desert *Chlorella*. Desert *Chlorella* grows in an extremely arid environment, with a maximum temperature of 67.2 °C in intense heat and a temperature difference of more than 40 °C between day and night. The winter is cold, with low precipitation and high evaporation (*Zhang et al., 2020*). *Lewis & Flechtner (2002)* found that desert green algae evolved from aquatic green algae at least five independent times. Therefore, desert *Chlorella* has adapted to the extremely harsh arid environment, making it an excellent material for studying the molecular biology of stress resistance at the single-cell level (*Mu et al., 2016*). In the past, studies of desert algae have focused mainly on the algal composition of microbiotic crusts and the characteristics and functions of crust formation (*Ashley & Rushforth, 1984*; *Garcia-Pichel, Lopez-Cortes & Nubel, 2001*; *Lewis & Flechtner, 2002*; *Zhao, Zhang & Zhang, 2006*). In recent years, *Li et al. (2012)* and *Wang et al. (2014)* analyzed the phylogeny of several strains of microalgae in the Gurbantonggut and Taklimakan Deserts. Among these algae, GTD7c-2, TLD2A, TLD6B are classified as *Chlorella*. *Gong et al. (2013)* compared different cultivation methods for desert *Chlorella* GTD8A1. *Zhu et al. (2014)* and *Mu et al. (2016)* measured the oil content of desert microalgae and desert *Chlorella*, respectively, and made a preliminary assessment of their potential for conversion to biodiesel.

Previous studies on lipid accumulation in freshwater *Chlorella* under salt stress have provided useful information, but there have been no reports on the mechanisms of lipid accumulation in desert *Chlorella* under salt stress. Desert *Chlorella* sp. TLD6B differs from fresh water algae, as it grows in extremely harsh environments. We must first clarify the mechanistic relationship between salinity stress and lipid accumulation before we can further develop desert *Chlorella* sp. TLD6B for use in biofuel production. Studies have shown that lipid accumulation in *Chlorella* can be induced by salt stress (*Wang et al., 2016*) and that salinity enhances microalgal lipid accumulation for biofuel production (*BenMoussa-Dahmen et al., 2016*; *Srivastava, Nishchal & Goud, 2017*). High salinity is an important factor that stimulates lipid accumulation, but it also has adverse effects on biomass growth and lipid production in microalgae (*Markou & Nerantzis, 2013*; *Ho et al., 2014*). *Sharma & Chauhan (2016)* used comparative transcriptomics to identify the molecular components associated with differences in lipid accumulation between the microalgal species *Scenedesmus dimorphus* and *S. quadricauda*. *Gao et al. (2014)* used

transcriptomics, genomics, and proteomics to characterize lipid accumulation mechanisms of the oleaginous microalga *C. protothecoides*.

Here, we studied *Chlorella* sp. TLD6B, a desert *Chlorella* collected from the Taklimakan Desert that exhibits greater drought and salt tolerance than freshwater *Chlorella* (unpublished results). We investigated the mechanism of lipid accumulation in *Chlorella* sp. TLD6B under salt stress by measuring relevant physiological parameters and analyzing whole-genome gene expression profiles under different levels of salt stress. This research aims to clarify the relationship between salt stress and oil accumulation in *Chlorella* sp. TLD6B and to promote its development and utilization in biofuel production.

## MATERIALS AND METHODS

### Algal strain and culture conditions

The experimental algae were collected from the Taklimakan Desert (37°36′N, 80°23′E, 1,254 m altitude, 3.5 °C average surface temperature, 0.4 M average NaCl salt concentration, 71.2 W m$^{-2}$ net radiation at 16:00) in Xinjiang Province, China. Field experiments were approved by the College of Life Sciences of Shihezi University (project number: 2011-GB-200900-G). The algae were isolated in the laboratory from a mixed culture of Taklimakan Desert soil samples and were identified as *Chlorella* sp. TLD6B (*Wang et al., 2014*). They were maintained in an autotrophic culture system at the College of Life Sciences, Shihezi University, China. Bold basal medium (BBM; *Nichols, 1973*) was autoclaved, and algal cells in the logarithmic growth phase were inoculated into Erlenmeyer flasks containing 500 mL BBM medium at a ratio of 20% (V/V). The initial cell density was controlled at an OD$_{680}$ of approximately 0.1, and the preparation of BBM medium is described in Table S1. Light was provided by white fluorescent lamps at an intensity of 72 μmol photons m$^{-2}$ s$^{-1}$ with a 12h/12 h light-dark cycle, and the temperature was 23 °C. Flasks were shaken three times a day at regular intervals and cultured for 24 d (*Mu et al., 2016*).

### Salt stress treatments

Algal cells in the logarithmic growth phase were inoculated into Erlenmeyer flasks containing 300 mL BBM medium, and 0.0, 0.1, 0.2, 0.4, 0.6, or 0.8 M NaCl was added to supplement the medium to a final volume of 500 mL. The 0.0 NaCl treatment was the control, and there were three replicates of each treatment. Algal culture was performed as described above. Samples were obtained every 6 d after inoculation to measure physiological and biochemical indexes such as biomass, carbohydrate concentration, and total lipid concentration.

### Measurement of physiological and biochemical indexes

Culture solution (100 mL) was added to 10-mL centrifuge tubes at 4 °C, and the tubes were centrifuged at 8,000 rpm for 15 min. The supernatant was discarded, and two mL of deionized water was added. The cell sediments were washed three times by vortexing (5–10 s each time), then divided into two equal parts. One part was used for dry weight measurement, and the other was immediately frozen in liquid nitrogen and stored at −80 °C for further analysis.

Optical density at 680 nm ($OD_{680}$) was determined using a UV-5800H UV–VIS spectrophotometer (Metash, Shanghai, China), and a growth curve was drawn (*Gong et al., 2013*). The centrifuged *Chlorella* sp. TLD6B sediment was dried at 80 °C for 24 h to measure the dry weight (*Wan et al., 2015*). The specific growth rate was calculated using the method of *Wang et al. (2016)*.

$$\mu = \frac{\lg N_1 - \lg N_0}{t_1 - t_0} \times 3.322 \tag{1}$$

where $\mu$ is the specific growth rate, mg $L^{-1}$ $d^{-1}$; $N_0$ is the biomass from the previous sampling date, mg $L^{-1}$; $N_1$ is the current biomass, mg $L^{-1}$; and $t_0$ and $t_1$ are the culture times, $t_1 - t_0 = 6$ d.

The phenol-sulfuric acid method was used to measure carbohydrate content (*Dubois et al., 1956*). The absorbance was measured at 490 nm on an iMark Microplate Reader (BIO-RAD, USA). Different concentrations of glucose were used to create a standard curve ($y = 8.55x - 0.07$, where x is the standard concentration ($\mu g \, mL^{-1}$) and y is the absorbance value), from which the glucose concentration $X_C$ of the sample could be calculated. $X_C$ ($\mu g \, mL^{-1}$) was defined as shown in equation Eq. (2), and carbohydrate content (CC, %) was defined as shown in Eq. (3).

$$X_C = \frac{(\triangle A + 0.07)}{8.55} \tag{2}$$

$$CC = \frac{X_C \times V \times 100\%}{V_C \times W} \tag{3}$$

where $\triangle A$ is the absorbance at 490 nm, V is the total volume of the sample (mL), $V_C$ is the volume of the subsample used for measurement ($\mu L$), and W is the dry weight of the sample (g $L^{-1}$).

The lipid content was measured by the colorimetric sulfo-phospho-vanillin method as described by *Mishra et al. (2014)* with some modifications (*Li et al., 2012*). The absorbance was measured at 528 nm, and the corresponding linoleic acid concentration ($\mu g \, mL^{-1}$) was calculated from the standard curve ($y = 0.0005x + 0.008$). $X_L$ ($\mu g \, mL^{-1}$) was defined as shown in Eq. (4), and the lipid content (LC, %) was defined as shown in Eq. (5).

$$X_L = \frac{(\triangle A - 0.008)}{0.0005} \tag{4}$$

$$LC = \frac{X_L \times V \times 100\%}{V_L \times W} \tag{5}$$

where $\triangle A$ is the absorbance at 528 nm, V is the total volume of the extract (mL), $V_L$ is the volume of the subsample used for measurement ($\mu L$), and W is the dry weight of the sample (g $L^{-1}$).

## RNA extraction and sequencing

Based on the results of physiological measurements under salt stress, we selected *Chlorella* sp. TLD6B cells grown under low salt stress (0.1 M NaCl addition) and high salt stress (0.8 M NaCl addition) for 18 d for use in transcriptome sequencing and analysis. Samples

collected from the control group, the 0.1 M NaCl treatment group, and the 0.8 M NaCl treatment group were labeled CK18, Nacl1, and Nacl2, respectively. Each treatment was replicated two times. Samples were centrifuged at 8000 rpm for 8 min at 4 °C. After removal of the supernatants, *Chlorella* sp. TLD6B cells were washed with distilled water three times and stored at −80 °C for RNA extraction.

Total RNA was isolated from all samples using the TRIzol reagent (Invitrogen, USA) according to the manufacturer's instructions. Total RNA was sent to Beijing Novogene Technology Co., Ltd. for RNA quality assessment, library construction, and sequencing (150-bp paired-end reads) on the Illumina NovaSeq 6000 platform. Raw sequence reads have been deposited into the NCBI GEO under accession number GSE162916.

## Transcriptome analysis

The total raw reads were processed using FASTQC software with default parameters. High-quality clean reads were obtained by removing reads that contained adapters, reads that contained more than 10% unknown nucleotides (N), and low-quality reads that contained more than 50% low-quality bases ($Q_{phred} \leq 20$). At the same time, the GC content, Q20, Q30, and sequence repetition level of the clean reads were calculated. Transcriptome de novo assembly was performed using Trinity v2.4.0 software (*Grabherr et al., 2011*) with the parameter Kmer = 25. Corset v1.05 (*Davidson & Oshlack, 2014*) was used with default parameters to cluster the transcript sequences in order to remove redundant sequences and obtain unigene sequence sets for subsequent analysis. The longest transcript of each subcomponent was used as the unigene for functional annotation. All assembled unigenes were finally annotated by comparing their sequences against the NR (NCBI non-redundant protein sequences, $e$-value = $10^{-5}$), NT (NCBI nucleotide sequences, $e$-value = $10^{-5}$), SwissProt ($e$-value = $10^{-5}$), GO (Gene Ontology, $e$-value = $10^{-6}$), KEGG (Kyoto Encyclopedia of Genes and Genomes, $e$-value = $10^{-10}$), PFAM (protein family, $e$-value = 0.01), and KOG/COG (euKaryotic Ortholog Groups/ Clusters of Orthologous Groups of Proteins, $e$-value = $10^{-3}$) databases. For NR, SwissProt, and KOG/COG annotations, Diamond v0.8.22 software was used. For NT annotation, Blastn (NCBI blast 2.2.28+) was used. PFAM protein family alignments were performed using the HMMER 3.0 package. GO annotations were obtained based on the annotation results from NR and PFAM using Blast2GO v2.5 software (*Götz et al., 2008*). The KEGG Automatic Annotation Server of KAAS was used for KEGG pathway analysis. The clean reads were aligned to the Trinity transcripts using Bowtie 2 with default parameters (*Langmead & Salzberg, 2012*). Following alignment, raw read counts for each transcript were derived using RSEM v1.2.15 (*Li & Dewey, 2011*) with default parameters and then normalized to FPKM (Fragments Per Kilobase of transcript per Million mapped reads) (*Trapnell et al., 2010*). The read counts were used as input for the DESeq R package (1.10.1) (*Anders & Huber, 2010*) to identify differentially expressed genes (DEGs) using the thresholds adjusted *P* value (padj) <0.05 and |log$_2$ FoldChange| ≥ 1. DEGs were mapped to terms in the GO and KEGG databases for functional and pathway analysis. The GOseq R package (*Young et al., 2010*) was used for GO enrichment analysis, and GO terms with corrected *P* value <0.05 were defined as significantly enriched in the DEG set. Significantly enriched
KEGG pathways were identified in the DEGs using KOBAS v2.0.12 with a corrected $P$ value <0.05 (*Mao et al., 2005*). The GO and KEGG enrichment maps were generated using the Beijing Novogene Technology Co., Ltd. server. The two figures were merged using Adobe Photoshop CS5 (Adobe Inc.; San Jose, CA, USA).

## Verification of RNA-seq gene expression by quantitative real-time PCR

Samples of total RNA (3 μg) were taken from the same total RNA used for RNA sequencing and used for the synthesis of first-strand cDNA with the M-MLV reverse transcription kit (Takara, Japan) according to the manufacturer's instructions. The cDNA obtained by reverse transcription was diluted 10-fold for use. Primer 5 software was used to design gene-specific real-time quantitative PCR (qRT–PCR) primers. qRT–PCR was performed using the Roche Light Cycler 480 system (Roche, Rotkreuz, Switzerland) and the SYBR Premix ExTaq kit (Takara, Japan) with a 10-μL PCR reaction system. The thermal cycle procedure was as follows: pre-incubation at 95 ° C for 5 min, followed by 40 cycles of 95 ° C for 15 s, 60 ° C for 15 s, and 72 ° C for 20 s. The $2^{-\Delta\Delta Ct}$ method was used to calculate relative gene expression using the *GAPDH* gene as a reference (*Livak & Schmittgen, 2001*). Pearson correlations between RNA-seq and qRT–PCR expression fold changes were calculated using SPSS 20.0. Treatment differences were assessed by analysis of variance using SPSS 20.0 (IBM Inc., Chicago, USA).

## Data analysis

All physiological data are presented as the mean ± standard deviation of three replicates. SPSS 20.0 was used to perform one-way analysis of variance (ANOVA) and correlation analysis, and Tukey's test was used to identify differences among individual treatments ($P$ < 0.05). Correlations between biomass and physiological indexes after different durations of salt stress were obtained using the bivariate module in the correlate analysis tool of SPSS 20.0. Graphs were constructed using SigmaPlot 12.5 software.

# RESULTS

## Effects of different salt stress levels on the physiological characteristics of *Chlorella* sp. TLD6B

Biomass increased under low salt stress and decreased under high salt stress, and *Chlorella* sp. TLD6B entered the logarithmic growth phase on the sixth day of culture. The $OD_{680}$ measurement increased rapidly before day 18, but slowed by day 24, entering a stable phase. The $OD_{680}$ measurement gradually decreased as the concentration of NaCl added to the medium increased from 0.1 M to 0.8 M (Fig. 1A and Table S2). The algae were able to tolerate high levels of salt (0.6 M and 0.8 M NaCl addition), but their biomass was very low (Fig. 1B and Table S2). Biomass was higher in the 0.1 M and 0.2 M NaCl addition treatments than in the control, but the $OD_{680}$ measurement decreased with increasing NaCl concentrations in the remaining treatments and also decreased with time. In general, compared with the control, the 0.1 M and 0.2 M NaCl addition treatments significantly inhibited the $OD_{680}$ but increased biomass. Biomass increased markedly after 12 days of NaCl stress in the 0.2 M NaCl addition treatment, whereas addition of 0.4–0.8 M NaCl significantly inhibited both $OD_{680}$ and biomass.

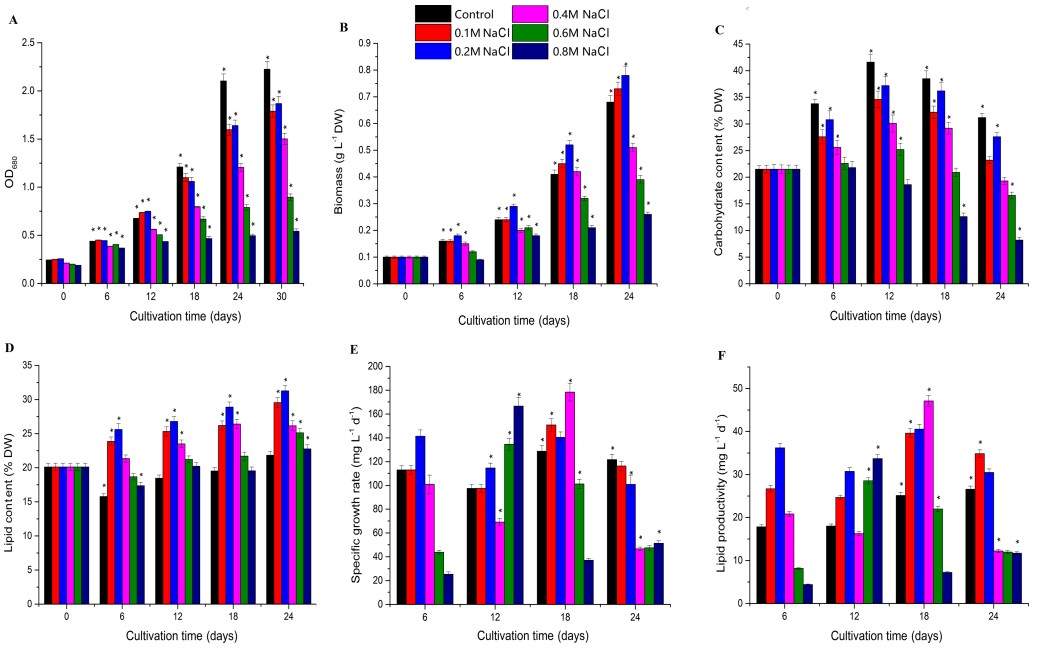

**Figure 1** **Effects of different levels of salt stress on the growth, biomass, carbohydrate content, lipid content, specific growth rate, and lipid productivity of *Chlorella* sp. TLD6B.** (A) Growth curves of *Chlorella* sp. TLD6B under different levels of salt stress. (B) Biomass of *Chlorella* sp. TLD6B under different levels of salt stress. (C) Carbohydrate content of *Chlorella* sp. TLD6B under different levels of salt stress. (D) Lipid content of *Chlorella* sp. TLD6B under different levels of salt stress. (E) Specific growth rate of *Chlorella* sp. TLD6B under different levels of salt stress. (F) Lipid productivity of *Chlorella* sp. TLD6B under different levels of salt stress. DW, dry weight. Bars represent SD.

Carbohydrate content decreased as NaCl concentration increased: with the exception of the 0.2 M NaCl addition treatment, the higher the NaCl concentration in the culture solution, the lower the algal carbohydrate content (Fig. 1C and Table S2). However, lipid content and biomass were improved by the addition of up to 0.2 M NaCl. Lipid content as a function of NaCl concentration was directly proportional to algal biomass, and *Chlorella* sp. TLD6B exhibited the highest lipid contents of 29.95% and 31.27% after the addition of 0.1 M and 0.2 M NaCl, respectively (Figs. 1B and 1D and Table S2). When NaCl concentrations greater than 0.2 M were added, lipid content decreased significantly as salinity increased (Fig. 1D and Table S2).

Carbohydrate content gradually increased early in cultivation (days 1–12) but decreased rapidly as total lipid content increased after day 12. At day 24, carbohydrate contents were 0.74-, 0.88-, 0.62-, 0.53-, and 0.38-fold that of the control in the 0.1, 0.2, 0.4, 0.6, and 0.8 M NaCl addition treatments, respectively (Fig. 1C and Table S2). By contrast, the lipid contents of the same treatments were 1.37-, 1.43-, 1.20-, 1.15-, and 1.13-fold that of the control (Fig. 1D and Table S2). At the same time, the stability of the specific growth rate of cells under low salt stress (0.1 and 0.2 M NaCl addition) was higher than that under medium and high salt stress (0.4–0.8 M NaCl addition), resulting in low lipid productivity

under high salt stress (Figs. 1E and 1F and Table S2). This is consistent with the change in lipid content.

The biomass of *Chlorella* sp. TLD6B was significantly correlated with its physiological parameters on day 18 of salt stress. Cell density ($OD_{680}$) and carbohydrate content were significantly correlated with biomass on day 18, but the correlation between biomass and total lipid content was highest ($r = 0.79$) (Table 1). Therefore, *Chlorella* cells under low salt stress (0.1 M NaCl addition) and high salt stress (0.8 M NaCl addition) on day 18 were selected for transcriptome sequencing to further study the mechanisms of lipid accumulation under salt stress.

## RNA-sequencing, de novo transcriptome assembly, and functional annotation

RNA-seq was performed on the mRNA extracted from the two NaCl treatments and the control, generating 963,078,184 raw reads. A total of 947,225,244 clean reads were obtained after filtering on base quality score and read length. The GC percentage of the clean reads was nearly 66.0%, and the Q20 was greater than 96% (Table S3). Trinity was used to create a de novo transcriptome assembly from the high-quality clean reads, producing 219,577 transcripts with an average length of 1,394 bp. 155,503 non-redundant unigenes were assembled; their length ranged from 200 to 23,825 bp with an average length of 1,842 bp (Tables S4 and S5).

All assembled unigenes were functionally annotated by searching against multiple public databases. Of the 155,503 unigenes, 111,238, 60,439, 84,173, 105,026, 109,098, and 51,484 were successfully annotated in the NR, NT, SwissProt, PFAM, GO, and KOG databases, respectively. Unigenes were also functionally annotated by searching against the KEGG database. A total of 47,032 (30.24%) unigenes had an annotated function in the KEGG database, and 131,623 (84.64%) unigenes were successfully annotated in at least one database (Table 2).

## DEGs in response to salt stress

There were a total of 16,402 DEGs across the low salinity (0.1 M NaCl addition, Nacl1) and high salinity (0.8 M NaCl addition, Nacl2) treatments, based on thresholds of $P < 0.05$ and $|\log_2 FC| \geq 1$ (Fig. 2A and Table S6). Of these, 1,326 were up-regulated and 2,304 were down-regulated under low salinity, and 9,776 were up-regulated and 5,798 were down-regulated under high salinity. Eight hundred twenty-eight unigenes were differentially expressed only under low salinity, 12,772 were differentially expressed only under high salinity, and 2,802 were differentially expressed under both conditions (Fig. 2B and Table S6). Of the 16,402 total DEGs, 82.9% were differentially expressed under only the low or the high salinity treatment.

## qRT–PCR verification of RNA-seq data

Nine DEGs in Nacl1 vs. CK18 (the 0.1 M NaCl treatment compared to the control on day 18) and Nacl2 vs. CK18 (the 0.8 M NaCl treatment compared to the control on day 18) were selected for qRT–PCR verification. These unigenes were mainly involved in fatty acid synthesis, TAG synthesis, the glycolysis/gluconeogenesis pathway, the pentose

**Table 1   Correlation coefficients between biomass and physiological parameters under salt stress of different durations.**

| Biomass under different stress durations | | OD$_{680}$ | Carbohydrate concentration | Total lipid concentration | Adding NaCl concentration |
|---|---|---|---|---|---|
| B_6 | Pearson correlation | 0.822* | 0.829* | 0.607 | −0.894 |
| | Significance (bilateral) | 0.045 | 0.041 | 0.201 | 0.016 |
| B_12 | Pearson correlation | 0.900* | 0.787 | 0.578 | −0.736 |
| | Significance (bilateral) | 0.014 | 0.063 | 0.229 | 0.096 |
| B_18 | Pearson correlation | 0.833* | 0.905* | 0.79 | −0.827 |
| | Significance (bilateral) | 0.040 | 0.013 | 0.061 | 0.042 |
| B_24 | Pearson correlation | 0.904* | 0.909* | 0.62 | −0.935 |
| | Significance (bilateral) | 0.013 | 0.012 | 0.189 | 0.006 |

**Notes.**
*$P < 0.05$. B_6 indicates the biomass on day 6, and so forth.

**Table 2   Gene functional annotation.**

| | Number of unigenes | Percentage (%) |
|---|---|---|
| Annotated in NR | 111,238 | 71.53 |
| Annotated in NT | 60,439 | 38.86 |
| Annotated in KEGG | 47,032 | 30.24 |
| Annotated in SwissProt | 84,173 | 54.12 |
| Annotated in PFAM | 105,026 | 67.53 |
| Annotated in GO | 109,098 | 70.15 |
| Annotated in KOG | 51,484 | 33.1 |
| Annotated in all databases | 23,373 | 15.03 |
| Annotated in at least one database | 131,623 | 84.64 |
| Total unigenes | 155,503 | 100 |

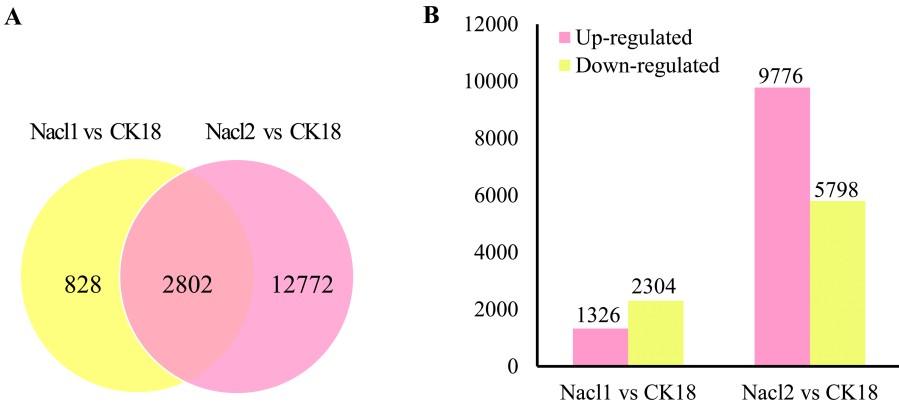

**Figure 2   DEGs in the Nacl1 (0.1 M NaCl addition) and Nacl2 (0.8 M NaCl addition) treatments.** (A) A Venn diagram of DEGs in the Nacl1 and Nacl2 treatments. (B) Numbers of up- and downregulated genes in the Nacl1 and Nacl2 treatments.

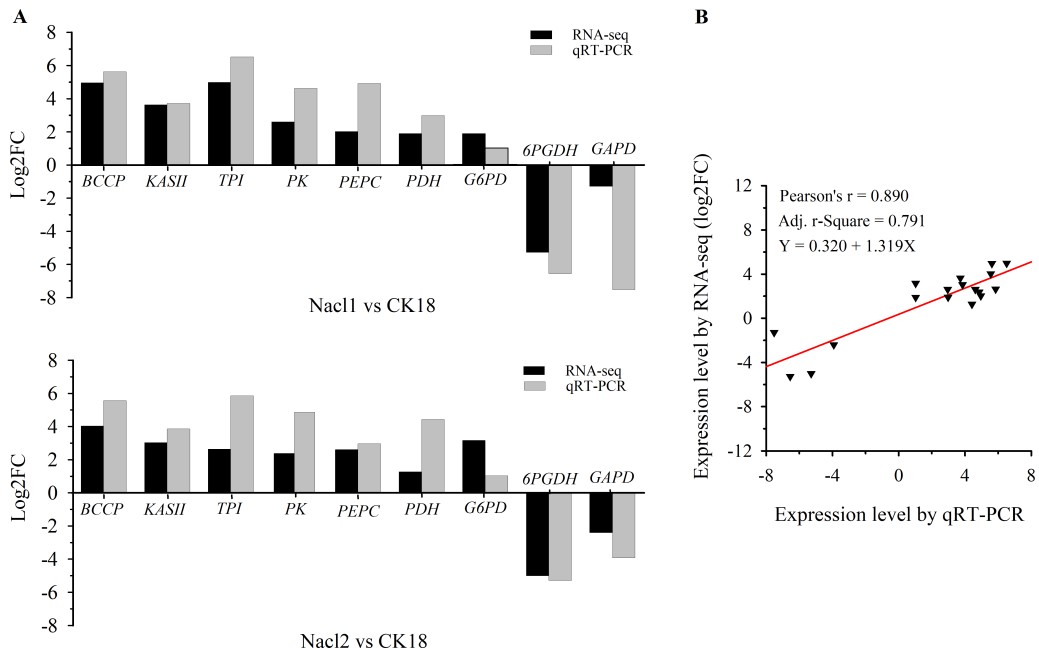

**Figure 3** **qRT–PCR verification of RNA sequencing results.** (A) Comparison of $\log_2$(fold change) values detected by RNA-seq and qRT–PCR. (B) Correlation between RNA-seq and qRT–PCR values. $\text{Log}_2\text{FC}$, $\log_2$ (fold-change) in gene expression between two groups. BCCP, biotin carboxyl carrier protein of acetyl-CoA carboxylase; KAS II, 3-oxoacyl-[acyl-carrier-protein] synthase II; TPI, triose-phosphate isomerase; PK, pyruvate kinase; PEPC, phosphoenolpyruvate carboxylase; PDC/PDH, pyruvate dehydrogenase; G6PD, glucose-6-phosphate dehydrogenase; 6PGDH, 6-phosphogluconate dehydrogenase; GAPD, glyceraldehyde 3-phosphate dehydrogenase.

phosphate pathway, and energy metabolism of the mitochondrial respiratory chain. Seven were up-regulated and two were down-regulated. The qRT–PCR primers listed in Table S7 were used to verify the RNA-seq data (Table S7), and the results are shown in Fig. 3A and Table S7. The expression patterns of nine unigenes were similar in the qRT–PCR and RNA-seq data, although there was some deviation in fold change values between the two methods. The qRT–PCR results were highly correlated with the RNA-seq results ($r = 0.890$, $r^2 = 0.791$), indicating that the RNA-seq data were generally accurate and reliable (Fig. 3B).

## GO and KEGG pathway enrichment analysis of the DEGs

To further explore their functions, the DEGs from the two salt treatments were annotated with GO terms and KEGG pathways for functional and pathway enrichment analyses (Table S6). Among the top 30 most highly enriched GO terms in the 828 specific DEGs from the low salinity treatment, one molecular function (MF) GO term was significantly enriched (corrected $P$ value <0.05): transferase activity, transferring one-carbon groups (GO:0016741). Among the top 30 most highly enriched GO terms in the 2,802 common DEGs, 10 biological process (BP) terms and 20 MF terms were significantly enriched (corrected $P$ value <0.05). In the BP category, the three most significantly enriched GO terms were G protein-coupled receptor signaling pathway (GO:0007186), defense response

to other organism (GO:0098542), and response to external biotic stimulus (GO:0043207). The three most significantly enriched MF GO terms were peptidase activity (GO:0008233), peptidase activity, acting on L-amino acid peptides (GO:0070011), and endopeptidase activity (GO:0004175). The most strongly and significantly enriched BP GO term in the 12,772 specific DEGs from the high salinity treatment was transcription initiation from RNA polymerase II promoter (GO:0006367). The three most significantly enriched cellular component (CC) GO terms were RNA polymerase II transcription factor complex (GO:0090575), nuclear transcription factor complex (GO:0044798), and transcription factor TFIIA complex (GO:0005672). The most significantly enriched MF GO term was secondary active transmembrane transporter activity (GO:0015291) (Fig. 4 and Table S8). There were no significantly enriched KEGG pathways in the low salinity-specific DEGs. Among the top 20 KEGG pathways enriched in the common DEGs, seven were significantly enriched (corrected $P$ value <0.05): plant hormone signal transduction (ko04075), arginine and proline metabolism (ko00330), glutathione metabolism (ko00480), betalain biosynthesis (ko00965), sulfur metabolism (ko00920), cysteine and methionine metabolism (ko00270), and nitrogen metabolism (ko00910). The 12,772 high salinity-specific DEGs were significantly enriched in mismatch repair (ko03430), DNA replication (ko03030), nucleotide excision repair (ko03420), proteasome (ko03050), photosynthesis (ko00195), porphyrin and chlorophyll metabolism (ko00860), and galactose metabolism (ko00052) (Fig. 5 and Table S9). Consistent with the KEGG analysis, GO enrichment analysis showed that common DEGs under low and high salinity were involved primarily in stress response, defense response, and membrane structure.

## Transcriptional expression of unigenes involved in fatty acid and TAG biosynthesis pathways

### Fatty acid biosynthesis

Acetyl-CoA, the precursor for fatty acid synthesis in the cytosol, derives from pyruvate (PYR) produced in glycolysis by pyruvate kinase (PK). Genes encoding two rate-limiting enzymes in glycolysis, hexokinase (HXK) and PK, were up-regulated under both the high and low salinity treatments, and 6-phosphofructokinase (PFK) was up-regulated under the high salinity treatment (Fig. 6 and Table S10). These changes would have promoted the accumulation of acetyl-CoA and energy.

The expression of key genes of the de novo fatty acid synthesis pathway also changed in response to salinity treatment. For example, the acetyl CoA carboxylase (ACCase) genes *BC* and *BCCP* were up-regulated on day 18 under both salinity treatments. The *α-CT* gene (Cluster-31803.29741) was up-regulated 2.7-fold under high salinity (Fig. 7A and Table S11), and the malonyl-CoA:ACP transacylase (*MCTA*) gene (Cluster-31803.71595) was also up-regulated only under high salinity (Fig. 7B and Table S11). Two 3-oxoacyl-[acyl carrier protein] synthase II (*KAS II*) genes (Cluster-31803.17167 and Cluster-31803.96146) were up-regulated under both salinity treatments, but these differences were not significant (Fig. 7C and Table S11). One *ω*-3 fatty acid desaturase (*FAD*) gene (Cluster-31803.83020) was down-regulated 4.5-fold under high salinity, and the other *FAD* gene (Cluster-31803.96909) was up-regulated 1.5-fold under low salinity (Fig. 7D and Table S11). A steroyl ACP

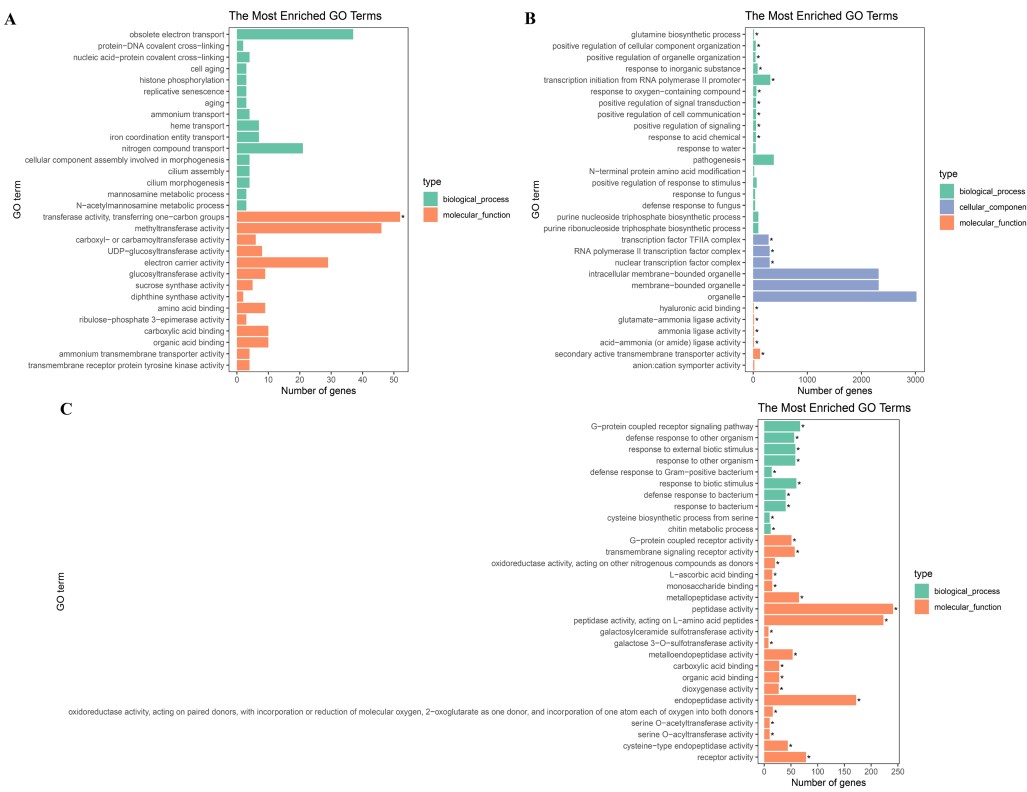

**Figure 4  GO enrichment of DEGs under salt stress.** (A) The top 30 most highly enriched GO terms in 828 unigenes specifically differentially expressed under low salinity (0.1 M NaCl addition); (B) the top 30 most highly enriched GO terms in 12,772 unigenes specifically differentially expressed under high salinity (0.8 M NaCl addition); and (C) the top 30 most highly enriched GO terms in 2,802 unigenes differentially expressed under both high and low salinity. The $x$-axis indicates the number of unigenes in each secondary GO classification. The $y$-axis indicates the secondary GO classification, and the three colors represent the three major types of GO terms (i.e., BP, CC, and MF). *corrected $p$-value $< 0.05$.

desaturase (*SAD*) gene (Cluster-31803.15683) was down-regulated under low salinity, whereas three *SAD* genes were up-regulated under high salinity (Cluster-31803.73844, Cluster-31803.73842, and Cluster-31803.73850) (Fig. 7E and Table S11).

### TAG biosynthesis

Four NAD-3-phosphate glycerol dehydrogenase (*GPD*) genes and one glycerol kinase (*GK*) gene were significantly up-regulated under high salinity. A gene encoding the glycolysis enzyme triose-phosphate isomerase (TPI) was up-regulated under both salinity treatments, which may have resulted in the accumulation of DHAP (Figs. 6, 8A, 8B, and 8C and Table S12) and promoted the synthesis of TAG. One *DGAT1* gene and seven *DGAT2* genes were identified in our transcriptome data. All but two were up-regulated (Cluster-31803.65991 and Cluster-31803.86613) (Figs. 6, 8D and Table S12).

## Expression of key unigenes related to lipid synthesis pathways

To further explore mechanisms of lipid accumulation in *Chlorella* sp. TLD6B under salt stress, we compared the expression of genes associated with the metabolic pathways of

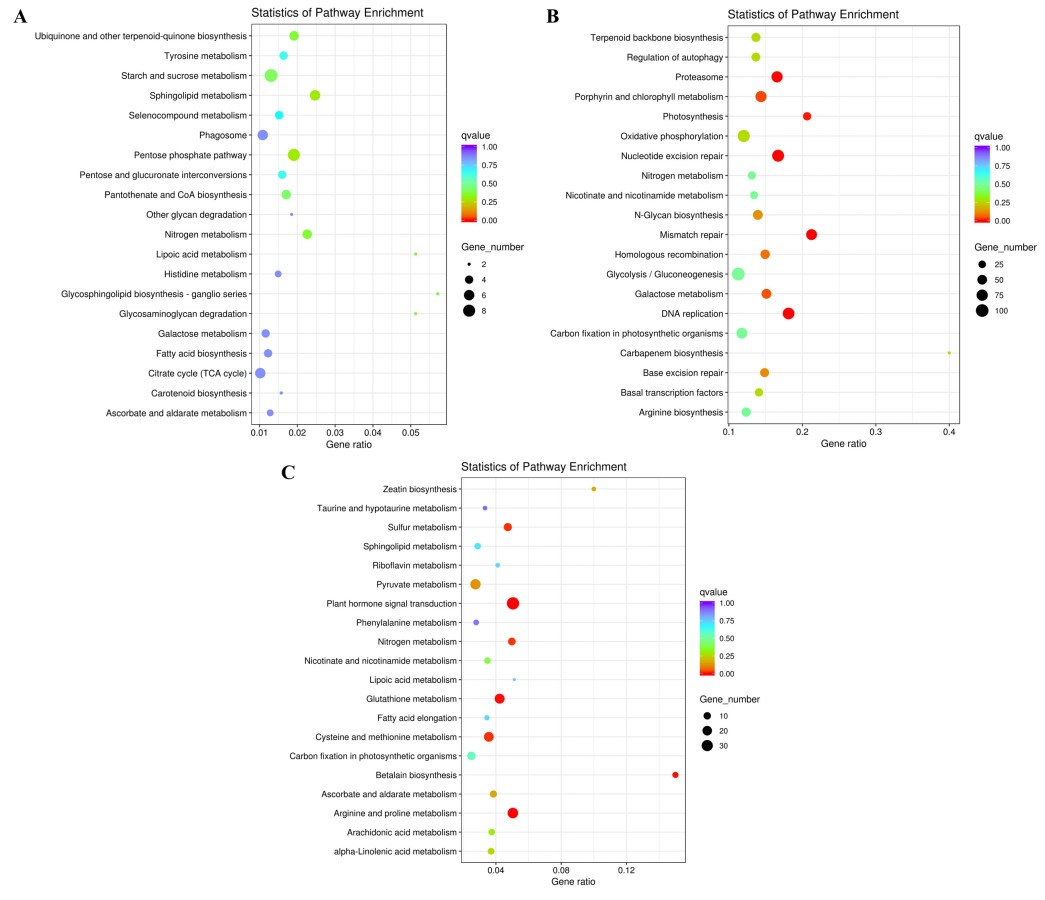

**Figure 5** **KEGG enrichment of DEGs under salt stress.** (A) The top 20 most highly enriched KEGG pathways in 828 unigenes specifically differentially expressed under low salinity (0.1 M NaCl addition); (B) the top 20 most highly enriched KEGG pathways in 12,772 unigenes specifically differentially expressed under high salinity (0.8 M NaCl addition); and (C) the top 20 most highly enriched KEGG pathways in 2,802 unigenes differentially expressed under both high and low salinity. The names of the pathways and their enrichment factors are shown. The size of the dot indicates the number of genes in a given pathway, and the color of the dot indicates the value of $-\log10$(corrected $p$ value).

starch-lipid biosynthesis. Under high salinity, a single gene encoding the starch-degrading enzyme $\alpha$-amylase (*AMY3*, Cluster-31803.62982) was up-regulated 4.6-fold (Fig. 6 and Table S10).

Furthermore, genes encoding ACL, PDC/PDH, ME, G6PD, and PEPC were primarily up-regulated. All of these genes showed greater expression under high salinity than under low salinity (Fig. 6 and Table S10). The upregulation of these genes whose products function in central carbon metabolism may directly or indirectly promote the accumulation of fatty acids.

## DISCUSSION

High salinity is generally not conducive to the growth and development of organisms because of Na$^+$ toxicity, excessive ROS accumulation, and difficulties related to osmotic

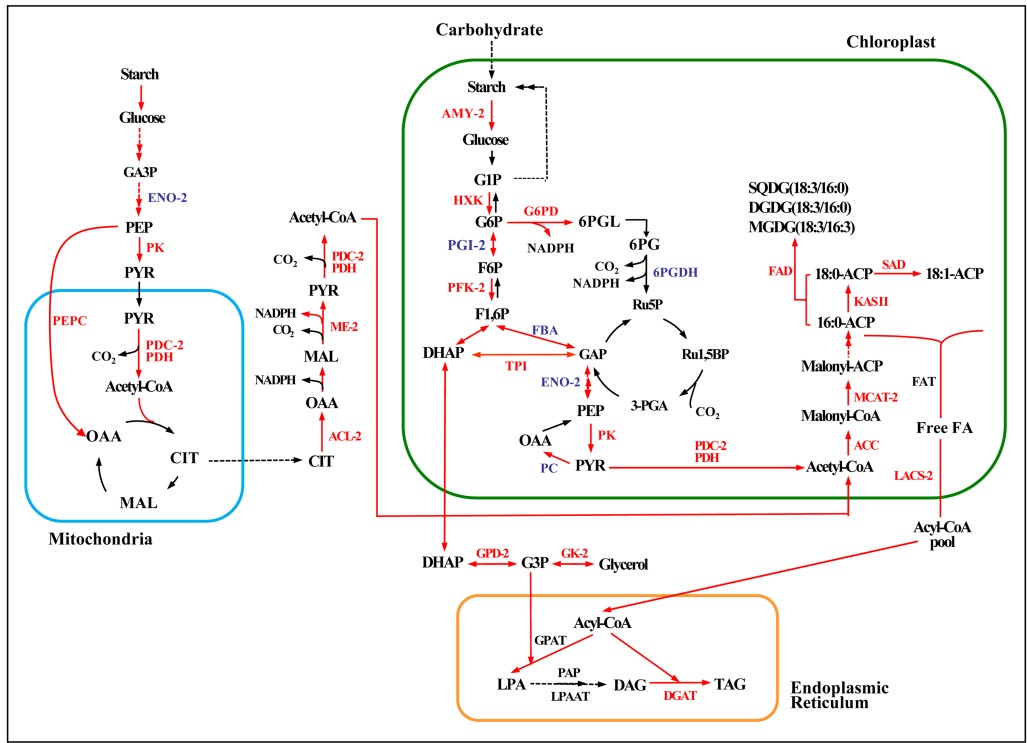

**Figure 6** **Enzymatic pathways of fatty acid and TAG biosynthesis and the expression of associated genes in response to two levels of salt treatment.** Arrows indicate enzymatic reactions, and red arrows indicate upregulation of the associated genes. Names of key lipid metabolism enzymes whose genes were upregulated are given in red, names of enzymes whose genes were associated with lipid metabolism are given in blue, and names of enzymes whose genes were not differentially expressed are given in black. "-2" after an enzyme name indicates that its gene was differentially expressed under high salt stress (0.8 M NaCl addition). Enzymes without this designation were differentially expressed under both high and low (0.1 M NaCl addition) salt stress treatments. AMY, α-amylase; HXK, hexokinase; PFK, 6-phosphofructokinase; DHAP, dihydroxyacetone phosphate; GAP, 3-phosphoglyceraldehyde; PYR, pyruvate; G3P, glycerol-3-phosphate; GPD, glycerol-3-phosphate dehydrogenase; GK, glycerol kinase; ACC, acetyl-CoA; ACL, ATP citrate lyase; ME, malic enzyme; LACS, long-chain acyl CoA synthetase; MCAT, malonyl-CoA:acyl-carrier protein transacylase; SAD, stearoyl ACP desaturase; FAD, ω-3 fatty acid desaturase; DGAT, diacylglycerol O-acyltransferase; TAG, triglyceride. Additional abbreviated gene names are defined in Fig. 3. Gene expression levels are shown in Figs. 7, 8 and Table S10.

balance and water absorption. The salt tolerance mechanisms of different microalgae may differ. To date, many studies have shown that salt treatment strongly promotes lipid synthesis in microalgae but also inhibits cell growth (*Sharma, Schuhmann & Schenk, 2012*; *Fan et al., 2014*; *Kato et al., 2017*; *Li et al., 2018*). According to current research, marine microalgae have higher salt tolerance than other algae such as *Dunaliella salina* (*Takagi, Karseno & Yoshida, 2006*; *BenMoussa-Dahmen et al., 2016*). By analyzing salt resistance and lipid content, we found that *Chlorella* sp. TLD6B could tolerate high levels of salt stress (0.8 M NaCl addition) that are lethal to many other freshwater algae (*Li et al., 2018*). *Chlorella* sp. TLD6B showed greater lipid accumulation following addition of 0.1 and 0.2 M NaCl, but this effect decreased at higher salt levels. This result indicates that *Chlorella* sp.

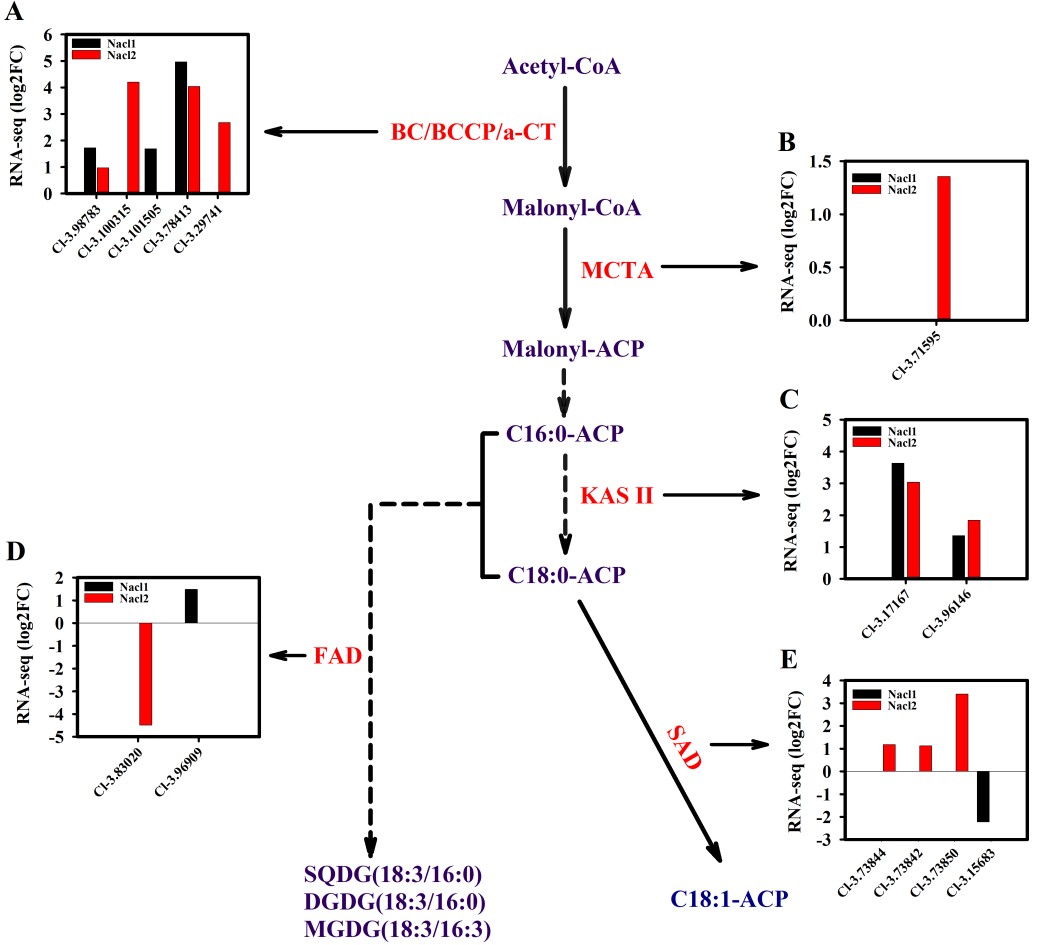

**Figure 7** Expression of *Chlorella* sp. TLD6B genes involved in the de novo synthesis of fatty acids pathway under low (0.1 M NaCl addition, Nacl1) and high (0.8 M NaCl addition, Nacl2) salt treatments. (A) Expression of BC/BCCP/ $\alpha$ -CT genes; BC, biotin carboxylase of ACCase; $\alpha$ -CT, carboxyltransferase $\alpha$ -subunit; (B) expression of MCTA gene; (C) expression of KAS II genes; (D) expression of FAD genes; (E) expression of SAD genes. Gene IDs such as Cluster-31803.100315 are abbreviated as Cl-3.100315. Additional abbreviated gene names are defined in Figs. 3 and 6.

TLD6B has a mechanism for increasing lipid accumulation under salt stress. By extension, the addition of 0.1 or 0.2 M NaCl can significantly promote *Chlorella* sp. TLD6B lipid accumulation and potentially reduce the production costs of biodiesel.

The salinity-induced accumulation of lipids in microalgae begins with a signal transduction cascade initiated by the perception of a salt stress signal. Downstream results include an increase in lipid and unsaturated fatty acid concentrations, the rapid accumulation of solutes that regulate cellular osmotic balance (soluble sugars, amino acids, and betaine), and the induction of the ROS scavenging system (*Ramos et al., 2011*). Here, we found that many key genes involved in glycerol, fatty acid, and lipid metabolism were significantly up-regulated in response to different salinity treatments, suggesting that the salt tolerance of *Chlorella* sp. TLD6B is related to its lipid metabolism. Low levels of

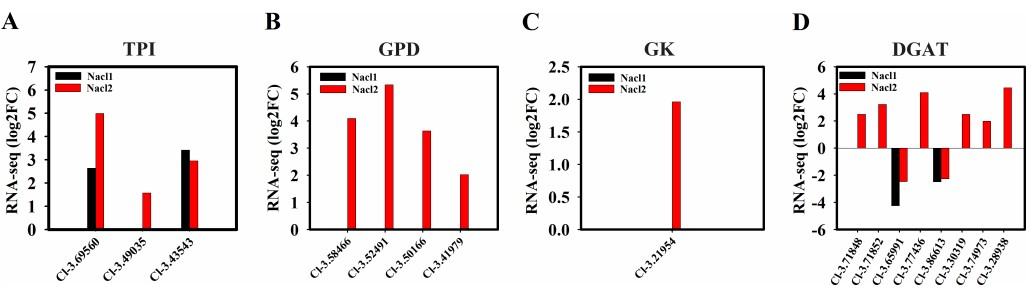

**Figure 8 Expression of *Chlorella* sp. TLD6B genes involved in the TAG biosynthesis pathway under low (0.1 M NaCl addition, Nacl1) and high (0.8 M NaCl addition, Nacl2) salt treatments.** (A) Expression of TPI genes; (B) expression of GPD genes; (C) expression of GK gene; (D) expression of DGAT genes. Gene IDs such as Cluster-31803.100315 are abbreviated as Cl-3.100315. Additional abbreviated gene names are defined in Figs. 3 and 6.

NaCl addition up-regulated the expression of genes related to fatty acid synthesis (*BC*, *BCCP*, *KAS II*) and promoted increased biomass and lipid production, indicating that low salinity is essential for the growth of *Chlorella* sp. TLD6B and ultimately promotes its lipid accumulation. Nonetheless, the regulation of lipid accumulation through the manipulation of salinity levels requires further study. *Rao et al. (2007)* reported that increased salt concentration in the growth medium altered the membrane lipid composition of *Botryococcus braunii* LB 572 cells: the proportion of stearic and linoleic acid decreased, and that of oleic acid rose. These results suggested that unsaturated fatty acids play a role in salt tolerance.

Previous studies have also shown that SAD is a key enzyme for the synthesis and metabolism of unsaturated fatty acids, directly determining the total amount of unsaturated fatty acids in vegetable oils as well as the ratio of saturated to unsaturated fatty acids (*Kachroo et al., 2007*). *FAD* is considered to be a candidate gene for the promotion of high oleic acid levels (*Jung et al., 2000a*). *Schwartzbeck et al. (2001)* used transgenic technology to inhibit FAD2 enzyme activity and increase the oleic acid content of corn oil from 25% to 64%. *Jung et al. (2000b)* increased the ratio of oleic acid to linoleic acid in peanuts by inhibiting FAD2 enzyme activity. Here, we also found that high salinity up-regulated *SAD* expression in *Chlorella* sp. TLD6B, whereas low salinity down-regulated *SAD* expression. One *FAD* gene (Cluster-31803.83020) was down-regulated 4.5-fold under high salinity, whereas another (Cluster-31803.96909) was up-regulated 1.5-fold under low salinity (Fig. 7D). This indicates that *SAD* and *FAD* may be involved in the regulation of fatty acid components in *Chlorella* sp. TLD6B. Low salinity may promote fatty acid synthesis, whereas high salinity may enhance unsaturated fatty acid synthesis, thereby reducing damage to cell membranes during salt stress (*Kan et al., 2012*). The lipid content of *Chlorella* sp. TLD6B increases under high salinity, and it is unknown whether the unsaturated fatty acid content also increases. Although we did not measure fatty acid composition, the transcriptome data indicated that *SAD* genes were up-regulated and an *FAD* gene was down-regulated under high salinity. This result suggests that the unsaturated fatty acid content may increase under high salt stress to reduce cell membrane damage.

Our results revealed that the pathway of lipid synthesis in *Chlorella* sp. TLD6B is finely regulated in response to different salinity conditions. First, upregulation of the key glycolysis genes *HXK* and *PK*, which implies enhancement of pentose phosphate metabolism, was likely to promote rapid acetyl-CoA accumulation and a stable intracellular energy supply. Therefore, high glycolysis rates may be one reason for the high lipid accumulation observed in *Chlorella* sp. TLD6B in response to NaCl addition. Second, ACCase catalyzes the formation of malonyl-CoA from acetyl-CoA and is the primary rate-limiting enzyme for de novo fatty acid synthesis. Upregulation of the corresponding gene may increase the rate of fatty acid synthesis (*Roesler et al., 1997*; *Zalutskaya et al., 2015*). We found that key genes of fatty acid synthesis, *ACCase* (*BC* and *BCCP*) and *KAS II*, were up-regulated after the addition of NaCl (Figs. 7A, 7C), which may also have enhanced fatty acid synthesis. Moreover, *ACL*, *PDC/PDH*, and *ME* in the PDH bypass were up-regulated, which may have accelerated the accumulation of precursor substrates and energy for fatty acid biosynthesis (*Tan & Lee, 2016*). The presence and high expression of these genes suggest that *Chlorella* sp. TLD6B uses a specific approach to ensure the rapid biosynthesis of fatty acids and TAG. Finally, the fatty acyl-CoA produced by de novo synthesis is esterified to produce TAG. The synthesis of TAG requires not only fatty acids but also glycerol 3-phosphate (G3P) (*Ohlrogge & Browse, 1995*). G3P is synthesized from glycerol by glycerol kinase (GK) or from DHAP by NAD-3-phosphate glycerol dehydrogenase (GPD) (*Yao et al., 2014*; *Huang et al., 2016*) (Figs. 6, 8B and 8C). In addition, research has shown that the overexpression of *GPDH* in microalgae promotes the conversion of DHAP to G3P (*Yao et al., 2014*) and that overexpression of fatty acid synthesis genes such as *ACCase* increases lipid accumulation (*Courchesne et al., 2009*). Both of these pathways compete for precursor substrates, and the flux of carbon out of glycolysis towards glycerol formation can cause cell growth to be inhibited. In our transcriptome data, four *GPD* genes and one *GK* gene were significantly up-regulated under high salt stress, and two *TPI* genes were up-regulated under both low and high salinity; these changes are beneficial to TAG synthesis but may inhibit cell growth. Therefore, salt stress only moderately increases the lipid concentration. This may explain why, in our study, carbohydrate levels gradually declined under salt stress while lipid accumulation increased. Moreover, the total lipid concentration under high salinity decreased to that of the control and was lower than the total lipid concentration under low salinity (Fig. 1D). Therefore, we suggest that salt stress alters the carbon flow between starch biosynthesis and lipid biosynthesis in *Chlorella* sp. TLD6B.

In addition, recent transcriptome studies have shown that microalgal lipid synthesis may be influenced by the conversion between starch and lipid synthesis (*Shang et al., 2016*; *Ho et al., 2017*). Here, only one *AMY3* gene (Cluster-31803.62982) was up-regulated 4.6-fold under high salinity. Lipid synthesis in *Chlorella* sp. TLD6B may therefore not be closely related to the conversion between starch and lipid biosynthesis. Its lipid accumulation may be controlled primarily by glycolysis and the synthesis of fatty acids and TAG.

## CONCLUSION

We performed physiological measurements and genome-wide transcriptome profiling of *Chlorella* sp. TLD6B under different levels of salt stress. *Chlorella* sp. TLD6B was able to

tolerate high salt stress (0.8 M NaCl addition). Lipid concentration first increased and then decreased with increasing levels of salt stress, reaching a maximum under the addition of 0.2 M NaCl. Comparative transcriptomic analysis revealed that salt stress enhanced the expression of genes related to sugar metabolism and fatty acid biosynthesis (the *ACCases BC* and *BCCP*, *KAS II*, and *GPDHs* involved in TAG synthesis), thereby promoting lipid accumulation under the addition of 0.2 M NaCl. However, high salt stress also inhibited cell growth. Expression of three *SADs*, whose encoded products function in unsaturated fatty acid biosynthesis, was up-regulated under high salinity (0.8 M NaCl addition). This research clarifies the relationship between salt tolerance and lipid accumulation, promoting the development and utilization of *Chlorella* sp. TLD6B for biofuel production.

## ACKNOWLEDGEMENTS

The authors would like to thank TopEdit for linguistic assistance during preparation of this manuscript.

### Funding

This study was supported by the National Natural Science Foundation of China (31460276) and the Shihezi University independently funded Project (ZZZC201840B). The funders had no role in study design, data collection and analysis, decision to publish, or preparation of the manuscript.

### Grant Disclosures

The following grant information was disclosed by the authors:
National Natural Science Foundation of China: 31460276.
Shihezi University independently funded Project: ZZZC201840B.

### Competing Interests

The authors declare that there are no conflict of interests.

### Author Contributions

- Hong Li conceived and designed the experiments, performed the experiments, analyzed the data, prepared figures and/or tables, authored or reviewed drafts of the paper, and approved the final draft.
- Jun Tan performed the experiments, prepared figures and/or tables, and approved the final draft.
- Yun Mu performed the experiments, analyzed the data, prepared figures and/or tables, and approved the final draft.
- Jianfeng Gao conceived and designed the experiments, authored or reviewed drafts of the paper, and approved the final draft.

## Field Study Permissions

The following information was supplied relating to field study approvals (i.e., approving body and any reference numbers):

Field experiments were approved by the College of Life Sciences of Shihezi University (project number: 2011-GB-200900-G).

## Data Availability

Chlorella sp. TLD6B transcriptome sequence data are available at NCBI GEO: https://www.ncbi.nlm.nih.gov/geo/query/acc.cgi?acc=GSE162916.

## Supplemental Information

Supplemental information for this article can be found online at http://dx.doi.org/10.7717/peerj.11525#supplemental-information.

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
