# Peer review of "Lipid accumulation of Chlorella sp. TLD6B from the Taklimakan Desert under salt stress"

_PeerJ, doi:10.7717/peerj.11525_

## Round 0.1 · original submission · Major Revisions

· Academic Editor

Major Revisions

The three reviewers expressed substantial concerns on the manuscript that should be all adequately addressed. The manuscript has to be corrected for the English language and preferably revised by an English specialist. There are a number of questions raised on the performed experiments and their designs. Also, revision of figures is required as suggested by the reviewers.

Reviewer 1 ·

Basic reporting

The manuscript is written clearly, however proof-editing is required.

Introduction refer to some relevant studies , but insufficiently covers recent discoveries on Chlorella species isolated from arid areas.

Article is well structured . Raw data are shared.

Experimental design

Research question is defined: how high salt concentration affect gene expression of the desert isolated of Chlorella. Physiological characterization included measurement on a few parameters.
Methods are well described.

Validity of the findings

The manuscript describes the effect of increasing NaCL concentration in the nutrient medium on the transcriptomic response of Chlorella sp. TLD6B.
Two concentrations were included in the transcriptomic study, designated as low (0.1) and high (0.8M).
Based on OD measurements , the growth was not substantially affected by 0.1M NaCl but was inhibited at 0.8 M NaCl.
Cultivation experiment rises several questions and concerns over selection of conditions.
The experiment duration is very long for the basic physiological experiment.
Considering long lag and long duration it is not clear when growth rates were determined.
The culture supplemented with high salt did not grow .
Total lipids increased insignificantly from the day 0 while carbohydrates severely decreased at 0.8 M.
What are other major components of biomass in this case?

Additional comments

Have you performed phylogenetic analysis of Chlorella sp. TLD6B?
Have you examined if cells are viable after a prolong exposure to salt stress?
The results indeed show that under high salinity level, the desert Chlorella sp. modulates gene expression of several fatty acid biosynthesis and TAG biosynthesis genes as well some genes of central carbon metabolism. And, the manuscript can be reorganized and rewritten to emphasize the effect on transcriptome. Based on the physiological data presented, the relevance to biodiesel production potential is questionable.

Reviewer 2 ·

Basic reporting

The novelty of this study lies in the fact that the authors have investigated the salt stress-related responses of Chlorella sp. in terms of gene expression and metabolic pathways using transcriptomic analysis coupled with physiological changes like that of biomass, sugar, and lipid content. This provides an insight into the salt-adaptation mechanisms of this microalgae thereby contributing to the basic knowledge of algal physiology. Despite the above-mentioned highlighting aspects, a lack of an overall clarity, structure, and coherence in several parts of the manuscript including introduction, methods, results and discussion significantly lowers the standard/quality of the manuscript.

1. The language of the manuscript needs to be improved substantially. Grammar- and vocabulary-related errors are quite frequently noted throughout the manuscript. For example, the phrases like “distributes”, “researches”, and “at home and abroad” in Lines 51-55 reflects poor standard of professional English.

2. The context of the study is quite unclear from the introduction. Thorough justification is required as to why we need to study the relationship between salinity stress and lipid accumulation in Chlorella sp. TLD6B. The authors have not provided sufficient references to present a clear background to their study especially in terms of highlighting the gaps in knowledge pertaining to the effect of high salinity on lipid accumulation in microalgae (maybe Chlorella in particular) and changes at the physiological and molecular/ gene level.

3. Figures:
a. Relevance: Table 1 and Figure 1 represent the same dataset with the table highlighting the significant differences between the relevant values. Significant differences are denoted by uppercase and lowercase letters which is quite confusing while trying to interpret the table. It is better to have the most prominent differences highlighted in the figure rather than the table.
b. High quality: The resolution of Figures 3-5 and 7 needs to be enhanced.
c. Labelling and description: Incorrect for y-axis in Figure 1C, legend labels inconsistent with color scheme in Figure 3, the different parts of the legend need to be consistent throughout all three bubble plots in Figure 4, statistical representations in the scatterplot of Figure 5 needs to be rectified.

4. Raw data in all supplied files have been presented in a clear and readable format. However, the files for Table 1 and Figure 1 provide the same information. This needs to be verified. Raw data supplied for Table 2 does not show how the coefficient of correlation was determined between biomass and other parameters keeping in mind different NaCl concentrations at each time point. Raw data file for Figure 5 does not include the negative values as represented in part A of figure for RT-qPCR results. The authors need to resolve these above-mentioned inconsistencies in the raw data provided.

Experimental design

1. The manuscript does demonstrate original research pertaining to the lipid accumulation in microalgae under salt stress conditions. The study fits well within the scope of the journal but, it is highly disappointing to note that the authors have not presented a clear research question that can resolve any potential gap in current knowledge. Examples are as follows:
Lines 47-50: The authors have presented one long and complicated sentence here which does not clearly indicate findings related to either lipid accumulation or lipid metabolism. The terms “lipid accumulation” and “lipid metabolism” have not been appropriately placed in the text and increases confusion to the reader throughout the introduction.
Lines 57-60: The sentence “After millions of years…” seems too hyped up. Please revise.
Lines 63: An introduction to the aspect of performing transcriptomic analyses of microalgae that can accumulate lipids would be more benefiting to highlight research gaps and enhance the overall standard of the manuscript.
Lines 66-68: This example of Chlamydomonas sp. seems irrelevant here especially when there are other references relating to Chlorella sp. itself.
Line 72: The authors state that their selected strain Chlorella sp. TLD6B from the desert has higher salt tolerance than freshwater Chlorella sp. based on their unpublished results. This is a strong statement considering that no evidence has been provided in this regard and that it is known that certain freshwater algae can tolerate slightly higher levels of salinity.

2. The description of the methods needs to be significantly improved in terms of the structure i.e. sequence of workflow, parameters assessed, data collection and analysis especially in the sections prior to RNA sequencing, transcriptome and gene expression analyses. Certain parts of the methods include irrelevant information while other parts of the methods require more technical details.

For example, in the first subsection of the methods i.e. “Algae strain and culture conditions”, in Lines 80-81, the authors strongly state that Chlorella sp. TLD6B was collected from the desert. This contradicts the latter half of the same sentence where an algal strain was isolated from a mixed environmental sample which was identified later as Chlorella sp. TLD6B. The environmental conditions such as temperature, salinity or light intensity during sample collection in the desert is missing here.
Lines 86-88: The authors need to provide detailed information on the type of light used in the laboratory. This can affect the physiology and lipid accumulation in algae. Other missing aspects here are the final salinity of the BBM medium Table S1 provided as a supplementary file, showed that 25 mg of NaCl was included during preparation. However, the original BBM medium composition contains 0.4M NaCl. Please clarify. Also the information pertaining to final volume in each flask per treatment is missing here.
Line 94: Usually analysis of algal physiology includes not only the biomass but also the morphological changes like changes in cell size/volume, changes in photosynthetic parameters, concentration of pigments, and changes in the mode of growth owing to salinity-stress in this context. Did the authors observe such differences in physiology in their culture?
Line 97: Please rectify the volumes mentioned for the tubes versus culture added.
Line 100: Can the authors explain why freezing in liquid nitrogen is necessary for dry weight measurements?
Lines 104-107: Brief descriptions of the method used to calculate values of Dcw% for carbohydrate and lipids is required for clarity.
Line 116: What do the authors mean by “ based on results of preceding studies…”? There is no reference cited here.
Lines 121-123: The authors have not provided any prior justification to why the drought-stressed Chlorella sp. TLD6B needs to be prepared here. This seems irrelevant here.
Line 156-157: Statistical analysis needs to include more details especially as to how correlation coefficients were determined between the physiological and biochemical indices.

Validity of the findings

The findings of this study have been very poorly presented i.e. lack of organization, and an overall coherent flow. The novelty of the study has also not been well discussed by the authors. It is incredibly difficult to understand clear trends from the physiological/biochemical datasets and the RNA-related datasets. For example,

Lines 162-192: The terms “cell growth rate”, “specific growth rates”, “cell growth” have been used interchangeably in several instances. Likewise, for the terms “carbohydrate consumption” versus “carbohydrate concentration” and “lipid concentration” versus “lipid productivity”.
The authors need to be aware of using the right terminology at each instance.

Lines 193-203: Significant correlations are highlighted well in Table 2 but not presented well in the text here. This paragraph needs to be more concise.

Line 232-234 and 242-244: Don’t these lines indicate the same message? It is better to avoid repetition. Likewise, for Lines 238-242 and Lines 247-252.

Line 254: The terms “NaCl1 VS CK18” and “NaCl2 VS CK18” have been introduced here for the first time. Please clarify.

Lines 310-311: The readers would benefit more from the description and/or comparison of different salt-tolerant mechanisms found in microalgae or different strains of Chlorella sp. itself.

Lines 317-318: This conclusion cannot be inferred from the presented findings since the investigated Chlorella sp. TLD6B was not compared to other microalgal strains in the same study and experimental conditions. Please revise.

Lines 323-324: What do the authors mean by “appropriate salt treatment…” ? If the highest lipid accumulation was observed at 0.2M NaCl, then this is the typical freshwater conditions ! Please verify. Also, how is salt treatment related to lowering the production cost of biodiesel?

Lines 379-380: It is probably the lack of clarity in the results and discussion but the authors need to be careful while mentioning that “salt stress could regulate carbon flow between starch biosynthesis and lipid biosynthesis in Chlorella sp. TLD6B”.

Reviewer 3 ·

Basic reporting

1. The manuscript contains some grammatical errors and needs to be proof read by a native/near native English speaking person. Some examples where the language could be improved include lines 47-50, 53-57, 313-314, 329-333 – the current phrasing makes
comprehension difficult.
2. There were many format problems throughout the manuscript, such as missing spaces in the word. For example, line 15: Chlorellahas; line 16: droughttolerance; line 17: beencultivated; line 23: andfatty; line 71: Itis; line 72: hasa; line 68: productivityin. Please check carefully.
3. Too many long and complicated sentences were included in the manuscript, such as line 47-50, line 325-329, line 329-333 and so on.
4. Some abbreviates were not explained, such as FPKM,DEGs, TAG.
5. The raw data of transcriptome should be deposited in the public database, such as NCBI.
6. The method needs more detail, such as the parameter for denovo assembly and annotation.
7. Figure 3 and 4 should be replaced by GO and KEGG enrichment of 2802 DEGs common shared by sample NaCl1 and NaCl2.
8. Figure 6: Genes and their transcriptional expression involved in fatty acid and TAG
biosynthesis pathways under different salt treatments. However, the expression level was not showed in the figure.

Experimental design

Authors mentioned that low salinity can promote the accumulation of lipid in Chlorella sp. However, why low salinity can promote the accumulation of lipid should be discussed. In other means, whether the accumulation of lipid would affect the salt tolerance of Chlorella sp. and how?

Validity of the findings

This research elaborated the relationship of salt tolerance
and lipid accumulation in Chlorella sp., which promotes the utilization of Chlorella sp. TLD6B.

Additional comments

no comment

---

## Round 0.2 · Major Revisions

· Academic Editor

Major Revisions

Both reviewers agreed that the paper has greatly improved, but raised a few significant issues. Please revise the manuscript after addressing all reviewer comments

Reviewer 1 ·

Basic reporting

This is a revised version of the manuscript. English has been substantially improved.

Experimental design

Please see comments to the authors.
Chlorella is a fast-growing species and generally, experiments can be perg\formes within several days, not weeks.

Validity of the findings

The validity of the transcriptomics results seems reliable.

Additional comments

The revised manuscript has been significantly amended, in particular in writing.
However, the authors have to better justify the study design (e.g. long lag and slow growth in control culture) and correct a few issues.
You note several times an increase in oleic acid. Where the data are shown?
389-390 (Jung et al., 2000a). Schwartzbeck et al. (2001) used transgenic technology to inhibit FAD
Fatty acid desaturases (FAD) are a group of enzymes with different substrate specificity. You shall not generalize by naming just FAD. Please be more specific.
You indicate certain molecular species of chloroplast glycerolipids in Fig. 7. Any data behind this presentation.

Reviewer 3 ·

Basic reporting

N/A

Experimental design

N/A

Validity of the findings

1. The accession number of transcriptomic raw data was not showed in th revised manuscript.
2. The conclusion number is different from abstract. Authors need to rephrase for the conclusion section.

Additional comments

This version of the manuscript has been greatly improved. However, some mistakes are still present in the manuscript.
1. Use italics for the names of Chlorella in the whole manuscript, such as line 56. Checked throughout the manuscript.
2. Line 60: delete surplus 'the'
3. The accession number of transcriptomic raw data was not showed in th revised manuscript.
4. The conclusion number is different from abstract. Authors need to rephrase for the conclusion section.
5. Where are legends for the supplemental materials?

---

## Round 0.3 · Major Revisions

· Academic Editor

Major Revisions

Thanks for addressing all the previous comments of the reviewers. Here are still a few points from the Section Editor to be addressed before making a final decision on your manuscript.

"The terms presented in figure 4, though done, are not readily available. The supplemental information should include the numerical GO: terms and the sequences associated with them. For instance, GO: annotations would help navigate enzymes mentioned in table for Suppl. Figs 7 and 8. There are also database terms to link to KEGG values. There are no sequences available for comparison as highlighted in Suppl. Table 3 and 4. As it is mentioned that there is no reference genome, it is important that representative sequences be presented to provide sequence diversity seen in clusters, and to have them partitioned based on the GO terms used. I cannot determine if the transcriptome data contained in the GEO resource is annotated properly to link sequences to assigned GO and KEGG terms. The manuscript is in general well written, but lacks details which add to the strength and utility of the manuscript. I will score this as requiring revisions until more information can assist the readers to navigate the data presented.

Journal manuscripts are often scanned by text-mining software that locates and extracts core data elements, like gene function. Adding standard ontology terms, such as the Gene Ontology (GO, geneontology.org) or others from the OBO foundry (obofoundry.org) can enhance the recognition of your contribution and description. This will also make human curation of literature easier and more accurate. None of this was visible."

EDITS
LINE NO: / BEFORE / AFTER / [COMMENTS]
LINE 131: / concent / content / [.]
LINE 143: / . / . / [not really a sentence; maybe “Eq. (3); where”]
LINE 158: / . / . / [not really a sentence; maybe “Eq. (5); where”]
LINE many: / upregulated / up-regulated / [.]
LINE many: / downregulated / down-regulated / [.]
LINE 465: / transcriptomes sequences / transcriptome sequence / [.]
LINE Suppl Raw Data Fig3: / . / . / [non-English notation]

---

## Round 0.4 · accepted · Accept

· Academic Editor

Accept

Thanks for the very thorough revision of the manuscript and for implementing the necessary changes as per the reviewers' comments.